# Anti-Inflammatory Effects of Adiponectin Receptor Agonist AdipoRon against Intervertebral Disc Degeneration

**DOI:** 10.3390/ijms24108566

**Published:** 2023-05-10

**Authors:** Hiroki Ohnishi, Zhongying Zhang, Takashi Yurube, Yoshiki Takeoka, Yutaro Kanda, Ryu Tsujimoto, Kunihiko Miyazaki, Tomoya Matsuo, Masao Ryu, Naotoshi Kumagai, Kohei Kuroshima, Yoshiaki Hiranaka, Ryosuke Kuroda, Kenichiro Kakutani

**Affiliations:** Department of Orthopaedic Surgery, Kobe University Graduate School of Medicine, Kobe 650-0017, Japan; o0717ooo@yahoo.co.jp (H.O.); yoshiki_tkk@hotmail.com (Y.T.); ykanda221@gmail.com (Y.K.); tsujiryu1105@yahoo.co.jp (R.T.); miya625819@gmail.com (K.M.); t.matsuo512@gmail.com (T.M.); smart_thomas0724@yahoo.co.jp (M.R.); kumagaiisumi19891107@gmail.com (N.K.); kohei.kuroshima@gmail.com (K.K.); yoshiagain@gmail.com (Y.H.);

**Keywords:** adiponectin, AdipoRon, intervertebral disc (IVD), disc degeneration, puncture model, spine

## Abstract

Adiponectin, a hormone secreted by adipocytes, has anti-inflammatory effects and is involved in various physiological and pathological processes such as obesity, inflammatory diseases, and cartilage diseases. However, the function of adiponectin in intervertebral disc (IVD) degeneration is not well understood. This study aimed to elucidate the effects of AdipoRon, an agonist of adiponectin receptor, on human IVD nucleus pulposus (NP) cells, using a three-dimensional in vitro culturing system. This study also aimed to elucidate the effects of AdipoRon on rat tail IVD tissues using an in vivo puncture-induced IVD degeneration model. Analysis using quantitative polymerase chain reaction demonstrated the downregulation of gene expression of proinflammatory and catabolic factors by interleukin (IL)-1β (10 ng/mL) in human IVD NP cells treated with AdipoRon (2 μM). Furthermore, western blotting showed AdipoRon-induced suppression of p65 phosphorylation (*p* < 0.01) under IL-1β stimulation in the adenosine monophosphate-activated protein kinase (AMPK) pathway. Intradiscal administration of AdipoRon was effective in alleviating the radiologic height loss induced by annular puncture of rat tail IVD, histomorphological degeneration, production of extracellular matrix catabolic factors, and expression of proinflammatory cytokines. Therefore, AdipoRon could be a new therapeutic candidate for alleviating the early stage of IVD degeneration.

## 1. Introduction

Low back pain (LBP) is a global health issue owing to a disability workforce that increases a socioeconomic problem, with a high lifetime prevalence of up to 85% and US medical expenditures of up to $100 billion/year [1,2]. Various factors such as unsanitary lifestyle, smoking, lack of physical activity, trauma, and exposure to mechanical load can cause LBP, and intervertebral disc (IVD) degeneration is recognized as a significant independent factor [3,4]. The IVD has a complex discoidal structure with the nucleus pulposus (NP) encapsulated by the annulus fibrous (AF) and endplates, resulting in the largest avascular organ in the body [5,6]. The AF comprises fibroblast-like cells and collagen type I, and the NP contains chondrocyte-like cells, proteoglycan, and water [7].

Various factors can also cause IVD degeneration, and obesity has primarily been implicated as a mechanical-risk factor [8]. In addition to its mechanical effects associated with being overweight, obesity also exerts metabolic and/or inflammatory effects on an IVD degeneration that is always accompanied by inflammation [9,10]. Inflammatory mediators and proinflammatory cytokines including tumor necrosis factor (TNF)-α, interleukin (IL)-1, IL-1β, IL-6, matrix-degrading enzymes such as matrix metalloproteinase (MMP)-3, MMP-13, and a disintegrin and metalloproteinase with thrombospondin motifs (ADAMTS)-4,5 have been reported to be involved in IVD degeneration [10,11,12,13]. The relationship between disc degeneration and inflammatory markers has been studied in recent years using a three-dimensional (3D) culture system owing to its superiority of retaining cellular characteristics and phenotype compared to those of conventional monolayer culture systems [14].

Adipose tissue is a large endocrine organ which secretes metabolic factors known as adipokines. Adiponectin, an adipokine, has been investigated for antidiabetic, antiatherogenic, and anti-inflammatory effects [15,16,17,18]. Interestingly, an inverse relationship has been noticed between adiponectin and obesity [19]. Regarding bone and cartilage diseases, adiponectin and its receptors, adiponectin receptor 1 (adipoR1), adipoR2, and T-cadherin have been reported to be involved in inflammation and degeneration. Adiponectin can upregulate tissue inhibitors of matrix metalloproteinases-2 (TIMP-2) and downregulate IL-1β-induced MMP-13 in human chondrocytes [20]. IVD tissues have several characteristics similar to those of cartilage, and NP cells have a chondrocyte-like phenotype [21]. The adiponectin receptors are widely expressed in both human and rat IVD cells, and TNF-α expression in rat NP cells is downregulated by adiponectin administration [22]. Similarly, TNF-α production by degenerated human NP cells is downregulated by adiponectin administration [23]. However, the clinical use of adiponectin as a drug has been reported to have significant disadvantages including high occurrence of adverse immunoreactions, need for constant intravenous injection of high doses to elicit effects, and the challenge of producing adiponectin on a large scale [24]. Therefore, as an alternative to adiponectin, AdipoRon, an agonist that can activate the adiponectin receptor, has been found to play biological roles similar to those of adiponectin [25]. Although AdipoRon has recently been applied in a variety of studies, little evidence exists regarding the effects of AdipoRon on IVD degeneration, particularly under proinflammatory conditions. Moreover, the effects of AdipoRon on other proinflammatory cytokines and extracellular matrix (ECM) metabolism have not yet been clarified.

We hypothesized that the administration of AdipoRon would decelerate IVD degeneration by downregulating proinflammatory cytokines. Therefore, this study aimed to elucidate the effect of AdipoRon on ECM metabolism, its anti-inflammatory potential, and the underlying mechanistic pathway in human IVD NP cells using an in vitro 3D culture system. Moreover, the decelerating effect of IVD degeneration of AdipoRon on rat tail IVD tissues was assessed using an in vivo animal model of degeneration induced by annular punctures.

## 2. Results

### 2.1. In Vitro IVD Experiments

#### 2.1.1. Effect of AdipoRon on Cell Viability

The viability of human NP cells was analyzed using a Cell Counting Kit-8 (CCK-8). AdipoRon at 5 μM concentration significantly decreased cell viability (versus control, 95% confidence interval (CI) 66.5–81.2). Based on these findings, 2 μM was selected as an effective but nontoxic concentration for subsequent experiments (Figure 1a).

#### 2.1.2. Effects of AdipoRon on the Expression of ACAN, COL2A1, TNF-α, IL-6, MMP-13, and ADAMTS-4

Real-time reverse transcription–polymerase chain reaction (RT–PCR) was performed to quantitatively analyze mRNA expression of anabolic and catabolic factors and proinflammatory cytokines. We analyzed ADAMTS-4 and MMP-13, representative catabolic factors of extracellular matrix which are recently attracting attention as potential therapeutic targets for IVD degeneration [26,27]. Human IVD NP cells were divided into four groups and treated with and without AdipoRon and/or IL-1β (control, group C; 2 μM AdipoRon, group A; 10 ng/mL IL-1β, group I; 2 μM AdipoRon + 10 ng/mL IL-1β, group A + I). The results indicated significantly lower expression of *TNFA* encoding TNF-α (*p* = 0.03), *IL-6* (*p* = 0.03), *MMP-13* (*p* < 0.01), and *ADAMTS-4* (*p* < 0.01) in group A + I than in group I; however, no significant differences in these parameters were observed between groups C and A. In addition, AdipoRon did not affect the anabolic factors (*ACAN* encoding aggrecan and *COL2A1* encoding collagen type II) (Figure 1b).

#### 2.1.3. Effects of AdipoRon on the Expression of TNF-α, IL-6, MMP-13, and ADAMTS-4

Multicolor immunofluorescence was performed to analyze the levels of IVD NP-associated notochordal CD24, catabolic factors, ADAMTS-4 and MMP-13, and proinflammatory cytokines TNF-α and IL-6 in AdipoRon-treated IVD NP cell clusters under IL-1β stimulation. Immunofluorescence analysis indicated significantly higher immunoreactivities of ADAMTS-4 (*p* < 0.01), MMP-13 (*p* < 0.01), TNF-α (*p* < 0.01), and IL-6 (*p* < 0.01) in group I than those in group C. In contrast, group A + I showed significantly lower immunoreactivities of ADAMTS-4 (*p* < 0.01), MMP-13 (*p* < 0.01), TNF-α (*p* < 0.01), and IL-6 (*p* < 0.01) than did group I. No significant difference in immunoreactivity of these catabolic factors and proinflammatory cytokines was observed between groups C and A (Figure 2).

Western blot analysis showed significantly higher expression of ADAMTS-4 (*p* < 0.01), MMP-13 (*p* < 0.01), TNF-α (*p* < 0.01), and IL-6 (*p* < 0.01) in group I than those in group C. Group A + I showed significantly lower expression of ADAMTS-4 (*p* < 0.01), MMP-13 (*p* = 0.02), TNF-α (*p* = 0.02), and IL-6 (*p* = 0.03) than did group I. However, no significant difference in the expression of these catabolic factors and proinflammatory cytokines was observed between groups C and A (Figure 3). TIMP-1 and TIMP-2 were less affected. These results suggest that AdipoRon exerts anti-inflammatory effects under proinflammatory conditions. Therefore, we further investigated the involvement of the adenosine monophosphate-activated protein kinase (AMPK)/nuclear factor kappa B (NF-κB) signaling pathways in the anti-inflammatory effect exerted by AdipoRon.

#### 2.1.4. Effects of AdipoRon on p65 Phosphorylation under Proinflammatory Conditions

AMPK pathway and NF-κB p65 at its downstream are known to be involved in controlling neuroinflammation and arthritis, and adiponectin activates the AMPK pathway [17,18,28,29]. We investigated the phosphorylation status of members of the AMPK pathway induced by AdipoRon with and without IL-1β stimulation in cultured human IVD NP cells. Western blot analysis showed increased level of p65 phosphorylation with a peak at 15 min after IL-1β supplementation (*p* < 0.01) (Figure 4a). After the 15 min stimulation, a significant reduction in p65 phosphorylation was noticed in group A + I compared to that in group I (*p* < 0.01), and a significant increase in AMPK phosphorylation was noticed in groups A (*p* < 0.01) and A + I (*p* < 0.01) compared to that in group C (Figure 4b,c).

### 2.2. In Vivo Experiments

#### 2.2.1. Effect of AdipoRon on the Height of Radiographic Disc in an Annular Puncture Model of Rat Tail IVD

Following the identification of anti-inflammatory characteristics of human IVD NP cells under AdipoRon treatment, we designed an in vivo study using a well-established annular puncture model of rat tail IVD characterized by its high accessibility and reproducibility [30]. Inflammatory cytokines are known to be elevated in rat puncture models, and these models are alternatives to NP cell experiments stimulated by IL-1β in vitro [30,31]. Discs were assigned into four groups: Group C was non-treated control; group A was injected with AdipoRon without annular puncture; group P with annular puncture was injected with phosphate-buffered saline (PBS); and group P + A with annular puncture was injected with AdipoRon. No significant difference was noticed between groups C and A. Group P showed a significantly lower disc height index (DHI) on days 14 (88.4 ± 2.3%, *p* < 0.01) and 28 (64.1 ± 6.9%, *p* < 0.01) after surgery than did group C. Group P + A also exhibited a significantly lower DHI on days 14 (88.9 ± 3.2%, *p* < 0.01) and 28 (74.1 ± 3.3%, *p* < 0.01) than did group C. However, DHI at postoperative 28 days was significantly higher in group P + A than in group P (*p* < 0.01) (Figure 5).

#### 2.2.2. Effect of AdipoRon on Histological IVD Degeneration in a Rat Model of Tail Puncture Model

Staining with safranin-O, fast green, and hematoxylin were performed to assess the general morphology of IVD with proteoglycan distribution. Control IVDs (group C) had round NPs at 14 and 28 days after starting the experiment, with a clear boundary observed between the AF and NP. Nucleated cells were evenly distributed in the NP area, with ECM proteoglycans organized into thin striations. No differences were observed between groups A and C. In contrast, in groups P and P + A, AF tissues presented a ruptured or tortuous pattern, a slight decrease in cell number, and slight condensation of ECM 14 days after puncture. Moreover, histologically significant degeneration was observed in groups P and P + A compared to that in groups C and A (C, 0.0 ± 0.0; A, 0.3 ± 0.5; P, 5.7 ± 1.4, *p* = 0.02; and P + A, 5.3 ± 1.1, *p* = 0.02). However, no significant difference was observed between groups P and P + A. After 28 days of puncture, degeneration was significantly more progressive in groups P and P + A than that in groups C and A (C, 0.4 ± 0.6; A, 0.6 ± 0.6; P, 12.0 ± 1.7, *p* < 0.01; and P + A, 7.6 ± 1.2, *p* < 0.01). Although each group exhibited progressive IVD degeneration after 28 days of puncture compared to that after 14 days of puncture, the histological grade in group P + A was significantly lower than that in group P (*p* = 0.02) (Figure 6).

#### 2.2.3. Effects of AdipoRon on the Expression of ECM Catabolic Factors and Proinflammatory Cytokines in a Rat Model of Tail Puncture

The in vivo expression of catabolic factors and proinflammatory cytokines was quantitatively analyzed by immunofluorescence. Immunopositivity was calculated as a percentage of DAPI-positive cells. In control, TNF-α-positive cells (C: 14 days, 20.8 ± 5.5%; 28 days, 27.1 ± 8.7%), IL-6-positive cells (C: 14 days, 19.8 ± 5.6%; 28 days, 26.5 ± 8.4%), ADAMTS-4-positive cells (C: 14 days, 23.6 ± 8.1%; 28 days, 23.1 ± 6.8%), and MMP-13-positive cells (C: 14 days, 19.6 ± 5.4%; 28 days, 20.4 ± 5.0%) were observed. Although a significant increase was noticed in the number of TNF-α-positive cells in groups P and P + A after 14 days of puncture, no significant difference was observed between groups C and A (A: 24.6 ± 5.1%, P: 44.1 ± 7.8%, P + A: 44.0 ± 8.5%). The numbers of IL-6-positive (A: 21.3 ± 5.4%, P: 41.1 ± 9.4%, P + A: 43.2 ± 6.7%), ADAMTS-4-positive (A: 23.7 ± 5.1%, P: 40.2 ± 6.1%, P + A: 41.6 ± 7.7%), and MMP-13-positive (A: 19.9 ± 6.8%, P: 40.6 ± 9.4%, P + A: 40.3 ± 4.5%) cells were consistent with TNF-α-positive cells (Figure 7a,b). The numbers of positive cells of each catabolic factor and proinflammatory cytokine in group P and P + A showed more increase after 28 days than after 14 days of puncture, whereas the increases in these catabolic factors and proinflammatory cytokines in group P + A were significantly suppressed compared to those in group P (TNF-α; *p* = 0.03, IL-6; *p* < 0.01, ADAMTS-4; *p* < 0.01, MMP-13; *p* = 0.003) (Figure 7a,b).

#### 2.2.4. Effects of AdipoRon on the AMPK Pathway in a Rat Model of Tail Puncture

The in vivo AMPK/NF-κB signaling pathway was analyzed by immunofluorescence. Immunopositivity was calculated as a percentage of DAPI-positive cells. In the control, a certain number of p-p65-positive cells was observed at both time points (C: 14 days, 22.2 ± 4.6%; 28 days, 27.4 ± 2.2%) (Figure 8a). However, after 14 days of puncture, a significant increase was noticed in the number of p-p65-positive cells in groups P and P + A, although no significant difference was observed between groups A and C (A: 20.6 ± 7.6%, P: 35.8 ± 5.4%, P + A: 34.9 ± 5.1%). After 28 days of puncture, the increase in group P + A was significantly suppressed compared to that in group P (*p* < 0.01) (Figure 8a). A certain number of p-AMPK-positive cells was observed at both time points in groups C and P (C: 14 days, 29.3 ± 7.5%; 28 days, 32.3 ± 7.0%, P: 14 days, 28.6 ± 5.9%; 28 days, 30.5 ± 6.4%) (Figure 8b). However, after 14 and 28 days of puncture, a significant increase was noticed in the number of p-AMPK-positive cells in groups A and P + A (*p* < 0.01), although no significant difference was observed between groups P and C (A: 14 days, 40.8 ± 6.0%; 28 days, 63.6 ± 10.4%, P + A: 14 days, 40.3 ± 5.5%; 28 days, 62.0 ± 7.6%) (Figure 8b).

## 3. Discussion

In this study, AdipoRon administration suppressed TNF-α expression in IVD cells, which is consistent with the results of previous studies [22,23]. The cause of IVD degeneration is not fully understood yet. Although numerous factors may initiate IVD degeneration, the imbalance of ECM metabolism, inflammatory response, apoptosis, and autophagy play essential roles in promoting the development of disc degeneration for molecular response [7,32,33]. In addition, mechanobiology, the new science that studies the interaction between mechanical stimuli and the biological behavior of cells and tissues, has been attracting attention in recent years as a key element in degenerative disc disease, and has led to the study of intervertebral disc degeneration [34]. TNF-α is a highly potent inflammatory cytokine and a pivotal contributor to IVD degeneration by participating in ECM degradation, inflammatory response, cell senescence, autophagy, and apoptosis [35,36]. Therefore, the inhibition of TNF-α may positively affect the suppression of IVD degeneration. Furthermore, TNF-α levels are closely associated with clinical conditions such as LBP and anti-TNF-α treatment can decrease the progression of disc degeneration and alleviate symptoms of LBP [22,35]. Based on these reports and our findings, suppressing TNF-α expression with AdipoRon would be an effective therapy to suppress IVD degeneration and alleviate LBP.

The effect of adiponectin on IL-6 production remains unclear. Adiponectin treatment increases IL-6 expression in synovial fibroblasts, endothelial cells, and osteoblasts in human [37,38]. IL-6 expression is not affected by adiponectin treatment in IVD cells [22]. In contrast, our results showed that AdipoRon administration suppressed IL-6 expression. The conflicting findings of IL-6 regulation by adiponectin might be owing to different in vitro experimental conditions. The importance of culturing cells in a 3D configuration that can retain cellular characteristics and phenotypes with more distinct interaction of cells with their ECM compared to that of traditional monolayer culture has been reported [39]. We conducted the in vitro experiments using a 3D culture system, TASCL, a two-layered device consisting of a lattice-like device and a membrane with microporous pores, which can generate many uniform cell clusters with a simple cell suspension. Although this device has not been extensively used in studies on IVDs, TASCL may have the advantage over the commonly used 3D culture system, alginate beads, that require complex procedures to obtain uniform cell clusters [40]. Furthermore, our in vivo data also showed the suppression of IL-6 expression by AdipoRon administration, which is consistent with our in vitro data.

We also demonstrated that the expression of MMP-13 and ADAMTS-4, ECM catabolic factors that are known to be associated with IVD degeneration, were suppressed by AdipoRon. No study has demonstrated that adiponectin suppresses the expression of these catabolic factors in IVD cells. Suppressing these catabolic factors can inhibit the progression of IVD degeneration [10,41]. Additionally, TNF-α promotes the induction of inflammatory cytokines such as IL-1β and IL-6 and ECM catabolic factors such as MMP-13 and ADAMTS-4 [33,42,43]. These reports support the idea that AdipoRon will contribute to suppressing IVD degeneration by modulating inflammatory cytokines and ECM catabolic factors. However, TIMP expression was not changed, suggesting that the imbalance between MMP-13 and TIMPs may promote IVD degeneration. In contrast, AdipoRon treatment did not show specific effects on ECM anabolic factors including aggrecan and collagen type II. Although adiponectin upregulates mRNA expression of genes encoding collagen type II, Sox9, and aggrecan in chondrocytes, the mechanism of action of adiponectin on ECM metabolism is still unclear [44]. Further investigation regarding the effect of adiponectin on ECM metabolism is necessary to elucidate the mechanisms of adiponectin action on IVD degeneration.

AdipoRon treatment increased AMPK phosphorylation with or without IL-1β stimulation and decreased p65 phosphorylation under IL-1β stimulation in IVD cells. Adiponectin has various physiological functions by initiating a series of tissue-dependent signal transduction events including phosphorylation of AMPK and p38 mitogen-activated protein kinase and inducing the activity of peroxisome proliferator-activated receptor alpha ligand. The activated NF-kB signaling pathway is involved in various inflammatory diseases, and various proinflammatory cytokines such as TNF-α and IL-1β activate the NF-κB signaling pathway [45]. In addition, AMPK pathway is involved in inhibiting NF-κB signaling and suppressing inflammation [46]. Taken together, our data suggest that AdipoRon treatment activates the AMPK pathway, which leads to the inhibition of NF-κB signaling, resulting in the anti-inflammatory effect in IVD cells under the inflammatory condition. Moreover, NF-κB has been reported as a key mediator of IVD degeneration; therefore, suppressing NF-κB signaling will inhibit IVD degeneration [47]. AdipoRon plays an important role in anti-inflammatory processes to protect IVD tissue from degeneration, suggesting that adiponectin is essential for IVD homeostasis and can be a therapeutic candidate for treating IVD degeneration.

We considered this degenerative model of rat puncture appropriate and valid for the present study because inflammatory cytokines are known to be elevated in puncture models [30,31]. However, the confounding effect of the level of spinal segments is unknown, which would need further investigation to be clarified. Our in vivo experiments showed consistent results with our in vitro data that AdipoRon administration suppressed proinflammatory cytokines and ECM catabolic factors, activated the AMPK pathway, and inhibited NF-κB signaling in human IVD degeneration. Furthermore, AdipoRon administration reduced the decrease in disc height and histological IVD degeneration, indicating that AdipoRon has a protective role against IVD degeneration.

Previous reports have neither evaluated the effects of AdipoRon under inflammatory conditions in vitro nor examined the effects in an in vivo degeneration model. Therefore, this study is highly novel with consistent results in vitro and in vivo, and may help support the safety and efficiency of AdipoRon for preventive use against IVD degeneration.

This study has several limitations. First, the study was conducted for IVD cells only under IL-1β stimulation in vitro; therefore, further studies under various conditions are necessary. Second, this study adopted a single concentration of AdipoRon in in vivo experiments. Finally, AdipoRon was administered concurrently with the puncture in this study; therefore, revealing the appropriate time for AdipoRon administration is necessary.

## 4. Materials and Methods

### 4.1. Ethics Statement

All human and animal experiments were performed under the approval and guidance of the Institutional Review Board (B210176) and Institutional Animal Care and Use Committee (P200807) at the Kobe University Graduate School of Medicine. Written informed consent was obtained from each patient following principles of the Declaration of Helsinki and laws and regulations of Japan.

### 4.2. Human IVD Experiments

#### 4.2.1. Tissues

Human IVD NP tissues were obtained from consenting patients, who underwent lumbar spine surgery for degenerative disease [*n* = 22: age, 50.2 ± 9.5 (16–70) years; male 11, female 11; Pfirrmann degeneration grade [48], median 3 ± 0.4 (2–4)] and were used to extract proteins.

#### 4.2.2. Cells

Immediately after surgery, human IVD NP cells were isolated and cultured from discarded surgical waste in patients with degenerative lumbar spine disease. IVD NP tissues were digested in Dulbecco’s Modified Eagle’s Medium (DMEM) (D5796; Sigma-Aldrich, St. Louis, MO, USA) with 10% fetal bovine serum (FBS; F2442; Sigma-Aldrich), 1% penicillin–streptomycin (26253-84; Nacalai Tesque, Kyoto, Japan), and 0.114% collagenase type II (LS004176; Worthington Biochemical, Lakewood, NJ, USA) for 1 h at 37 °C under 2% O_2_. The separated cells were cultured and grown up to approximately 80% confluence as a monolayer in DMEM with 1% penicillin–streptomycin and 10% FBS in an incubator (9000EX; Wakenyaku, Kyoto, Japan) at 37 °C under an atmosphere of 5% CO_2_ and 2% O_2_ to simulate a physiologically hypoxic IVD environment. In this study, we used only first-passage cells for evaluation to retain the phenotype.

#### 4.2.3. Assessment of Cell Viability

Cell viability was assessed using CCK-8 (CK04; Dojindo, Kumamoto, Japan) to identify the cytotoxicity of AdipoRon to NP cells. NP cells were cultured in DMEM with 10% FBS and 1% penicillin for 72 h at 37 °C. After three days of preculture, cells were trypsinized and seeded into 96-well plates at a density of 5 × 10^3^ cells per well with 100 µL DMEM and incubated at 37 °C for 24 h. Cells were then treated with various concentrations (0–20 µM) of AdipoRon (SML0998; Sigma-Aldrich) at 37 °C for 24 h, and 10 µL CCK-8 solution was added to each well and incubated for another 24 h. The optical density (OD) was measured at 450 nm using a microplate reader (BIO-RAD model 680, Hercules, CA, USA). Cell viability was calculated from the absorbance values (Figure 9a).

#### 4.2.4. Protein Extraction

Proteins were extracted from IVD NP tissues to evaluate protein expression in IVD NP cells by western blotting. Harvested tissues were homogenized using an MS-100R bead-beating disrupter (Tomy Seiko, Tokyo, Japan) for 30 s twice at 4 °C in a T-PER tissue protein extraction reagent (78,510; Thermo Fisher Scientific, Waltham, MA, USA) containing protease and phosphatase inhibitors [protease inhibitor cocktail (25955-11; Nacalai Tesque), phosphatase inhibitor cocktails 2 (P5726; Sigma-Aldrich), and phosphatase inhibitors cocktails 3 (P0044; Sigma-Aldrich)]. Finally, soluble proteins were collected by centrifugation at 20,000× *g* for 15 min at 4 °C. Protein concentration was determined using a bicinchoninic acid assay (23,227; Thermo Fisher Scientific). Samples were stored at −80 °C.

#### 4.2.5. RNA Isolation and RT–PCR

Cells were cultured in a monolayer medium, followed by seeding into a 3D culture system (TASCL, 600 well; Cymss-Bio, Tokyo, Japan) at a density of 1.0 × 10^6^ cells/TASCL to form cell clusters. After 48 h, cell clusters were treated with AdipoRon (2 μM, group A), IL-1β (10 ng/mL, group I) (ALX-520-001; Enzo Life Science, Farmingdale, NY, USA), or both (group A + I). Total RNA was extracted using a RNeasy mini kit (74104; Qiagen, Hilden, Germany), and RNA (0.1 μg) was reverse-transcribed using a High-Capacity cDNA Reverse Transcription Kit (4368814; Thermo Fisher Scientific). mRNA levels of the catabolic factors (*ADAMTS-4* and *MMP-13*) and proinflammatory cytokines (*TNFA* encoding TNF-α and *IL-6*) relative to *GAPDH* expression were analyzed by RT–PCR using SYBR Green (4367659; Thermo Fisher Scientific). Good feasibility of GAPDH as an endogenous control gene for disc cells was shown previously [49]. The primer sequences are listed in Table 1, which were obtained from published reports [50,51] and purchased from TaKaRa Bio (Shiga, Japan) and Thermo Fisher Scientific. Measurements were performed using an ABI Prism 7500 RT–PCR system (Thermo Fisher Scientific). Melting curve analysis was performed using the Dissociation Curves software (StepOne Real–Time PCR System Software version 2.3; Thermo Fisher Scientific) to ensure that only a single product was amplified. Relative mRNA expression was analyzed using the 2^−ΔΔCt^ method, and the value of vehicle control was set at 1.0. In quantitative RT–PCR analysis, the effects of treatment on target gene expression were calculated as relative values of the vehicle control. Samples of each patient were analyzed in duplicate, and single values were obtained by averaging the values of these replicates. Experiments were conducted six times using six patient samples (*n* = 6) (Figure 9a).

#### 4.2.6. In Vitro Immunofluorescence

NP cells were divided into four groups, cultured in TASCL microplates, and treated with and without AdipoRon and/or IL-1β (control, group C; 2 μM AdipoRon, group A; 10 ng/mL IL-1β, group I; 2 μM AdipoRon + 10 ng/mL IL-1β, group A + I). Cell clusters, formed on TASCL 48 h after treatment, were incubated with primary antibodies (1:200) against IVD NP-associated CD24 (sc-11,406; Santa Cruz Biotechnology, Santa Cruz, CA, USA), TNF-α (GTX110,520; GeneTex, San Antonio, TX, USA), IL-6 (GTX110,527; GeneTex), ADAMTS-4 (sc-16,533; Santa Cruz Biotechnology), and MMP-13 (LS-B3168; LifeSpan BioSciences, Seattle, WA, USA) overnight at 4 °C. Cell clusters were further treated with secondary antibodies conjugated with Alexa Fluor 488 (A21200; Thermo Fisher Scientific), Alexa Fluor 568 (A21468; Thermo Fisher Scientific) or Alexa Fluor 647 (A31573; Thermo Fisher Scientific) for 1 h at 25 °C. Finally, cells were counterstained with DAPI (D1306; Thermo Fisher Scientific). Immunofluorescent images of cell clusters on TASCL were obtained using a microscope (BZ-X700; Keyence, Osaka, Japan). Relative fluorescence intensities of 10 randomly selected cell clusters per image were measured using a microscope and BZ-X700 software as previously described [52] (Figure 9a).

#### 4.2.7. Western Blotting

NP cells were divided into four groups, cultured in TASCL microplates, and treated with and without AdipoRon and/or IL-1β (control, group C; 2 μM AdipoRon, group A; 10 ng/mL IL-1β, group I; 2 μM AdipoRon + 10 ng/mL IL-1β, group A + I). Equal amounts of protein (20 μg) were mixed with Laemmli buffer, boiled for 5 min, and loaded onto a 7.5–15% polyacrylamide gel (SDG-581; Bio Craft, Tokyo, Japan). Proteins separated using Tris (35434-21; Nacalai Tesque)–glycine (12-1210; Sigma-Aldrich)–sodium dodecyl sulfate buffer system were electroblotted onto a membrane, probed with 1:200–1:1000-diluted primary antibodies [anti-ADAMTS-4 (sc-16533; Santa Cruz Biotechnology), anti-MMP-13 (LS-B3168; LifeSpan BioSciences), anti-IL-6 (GTX110527; GeneTex), anti-TNF-α (GTX110520; GeneTex), anti-TIMP-1 (sc-5538; Santa Cruz Biotechnology), anti-TIMP-2 (5738; Cell Signaling Technology, Danvers, MA, USA) and anti-actin (A5441; Sigma-Aldrich)] for 12 h at 4 °C, and subsequently incubated with 1:2000-diluted secondary antibodies [anti-rabbit (NA934), anti-mouse (NA931); GE Healthcare, Chicago, IL, USA; anti-goat(sc-2020); Santa Cruz Biotechnology) for 1 h at 25 °C. Protein bands were visualized using enhanced chemiluminescence. Images were obtained using a Chemilumino Analyzer LAS-3000 Mini (Fujifilm, Tokyo, Japan). Band intensities were quantified using ImageJ (https://imagej.nih.gov/ij/, accessed on 17 August 2020).

#### 4.2.8. p65 and AMPK Phosphorylation Assay

We selected six time points (0, 15, 30, 60, 120, and 240 min) to determine the peak time of activation of the AMPK signaling pathway following the stimulation of cultured cells with 10 ng/mL IL-1β using western blotting. Furthermore, we cotreated the cultured cells with 2 μM AdipoRon and 10 ng/mL IL-1β and evaluated the effects of AdipoRon on the AMPK/NF-κB signaling pathway at the peak time point. In addition, western blotting was performed six times using six patient samples to evaluate AMPK phosphorylation. Primary antibodies against AMPK (AF6423; Affinity Biosciences, Cincinnati, OH, USA), anti-phospho-AMPK (Thr 172) (AF3423; Affinity Biosciences), anti-p65 (aAF5006; Affinity Biosciences), and anti-phospho-p65 (AF2006; Affinity Biosciences) were used. Representative immunoblots of six similar results are presented (*n* = 6) (Figure 9a).

### 4.3. Rat IVD Experiments

#### 4.3.1. Animals

The tails of twelve-week-old male Sprague–Dawley rats were used to develop a model of IVD degeneration [*n* = 18: weight, 408.1 ± 9.2 (398–417) g] [53]. Under general anesthesia, we used a 20-gauge (G) needle to puncture the skin of caudal vertebra to the center of NP at a depth of 5 mm with the handmade stopper previously reported [54]. The needle was rotated 360° and held in that position for 30 s before removal to induce more rapid degeneration [53,54]. Punctures at C10–C11 and C11–C12 were performed to assess the effects of AdipoRon, while the C8–C9 disc was later collected as the non-treated control (group C). At C9–C10, 2 μL AdipoRon was injected using a 33-G needle at the disc center through a 5 mm longitudinal skin incision (group A) to evaluate the effects of AdipoRon without puncture [55,56]. Immediately following the initial puncture, either vehicle (PBS; 2 μL per disc, group P) or AdipoRon (2 μM in 2 μL per disc, group P + A) was injected into 18 rats at the center of the IVD NP using a 27-G needle, which can pass through 20-G needle and is long enough to reach the center of IVD. (MS-N05; Ito Corporation, Shizuoka, Japan). The amount of AdipoRon administered per rat was set to 2 μM in 2 μL following a previous study [22,23,57,58]. Disc height evaluation, histomorphology, and immunofluorescence were performed 14 and 28 days after the puncture (Figure 9b).

#### 4.3.2. Radiography

Lateral radiographs were taken using a VPX-30E (Toshiba Medical Supply, Tokyo, Japan) system and IXFR film (Fujifilm) (exposure time, 40 s; distance, 40 cm; current, 3 mA; voltage, 35 kV). DHI was calculated by measuring, averaging, and normalizing the disc height with respect to the adjacent vertebral body heights in the anterior, middle, and posterior regions to correct the asymmetry form of each intervertebral disc. Disc height was measured using ImageJ (https://imagej.nih.gov/ij/, accessed on 22 November 2020), normalized to adjacent vertebral body heights as DHI, presented as the percentage of preoperative DHI [%DHI = (postoperative DHI/preoperative DHI) × 100), as previously described, and further normalized to the intact disc as normalized %DHI [normalized %DHI = (experimental %DHI/intact %DHI) × 100, *n* = 6 per group] (Figure 9c) [54].

#### 4.3.3. Paraffin-Embedded Tissue Preparation

Rat caudal functional spinal units (vertebral body–disc–vertebral body) were obtained after euthanasia. They were fixed en bloc in 4% paraformaldehyde (163-20145; Wako Osaka, Japan) for 1 day, demineralized in 10% ethylenediaminetetraacetic acid (345-01865; Wako) for 7 days, embedded in paraffin, and cut to obtain a mid-sagittal 7-μm section for histomorphology and immunofluorescence analysis.

#### 4.3.4. Histomorphology

Staining with safranin-O (S-0145; Tokyo Chemical Industry, Tokyo, Japan), fast green (10720; Chroma Gesellschaft Schmidt, Munster, Germany), and hematoxylin (30002; Muto Tokyo, Japan) were performed to visualize the morphological disruption of disc tissue. Images of the sections were captured using a BZ-X700 microscope and were graded by two blinded individual examiners from 0 (non-degenerative) to 16 (severely degenerative) for NP morphology, NP cellularity, NP–AF boundary, AF morphology, and endplate, according to a reported histological degeneration scale (*n* = 6 per group) [59].

#### 4.3.5. In Vivo Immunofluorescence

Multi-color immunofluorescence was performed to assess the levels of ECM catabolic factors, pro-inflammatory cytokines, and AMPK/NF-κB signaling pathway. After antigen-retrieval by Heat-Induced Epitope Retrieval with Dako Target Retrieval Solution (Dako, Glostrup, Denmark) at 90 °C for 20 min in the hot water dispenser and cooling at room temperature for 20 min, permeabilization, and blocking, sections were incubated with primary antibodies (1:200) against CD24 (sc-11406; Santa Cruz Biotechnology), ADAMTS-4 (sc-16533; Santa Cruz Biotechnology), MMP-13 (LS-B3168; LifeSpan BioSciences), TNF-α (GTX110520; GeneTex), IL-6 (GTX110527; GeneTex), AMPK (AF6423; Affinity Biosciences), phospho-AMPK (AF3423; Affinity Biosciences), p65 (AF5006; Affinity Biosciences), and phospho-p65 (AF2006; Affinity Biosciences) for 12 h at 4 °C, followed by incubation with secondary antibodies (1:400) conjugated with Alexa Fluor 488 (A-21200; Thermo Fisher Scientific), Alexa Fluor 594 (A-21468; Thermo Fisher Scientific), and Alexa Fluor 647 (A-31573; Thermo Fisher Scientific) for 1 h at 25 °C. DAPI (D1306; Thermo Fisher Scientific) was used for counterstaining. Images were captured using a BZ-X700 microscope (Keyence). The number of positive cells was counted in four high-power random fields (400×) using the BZ-X700 software. Average immunopositivity was calculated as a percentage relative to DAPI-positive cells (*n* = 6 per group).

### 4.4. Statistical Analysis

Statistical analysis was performed using IBM SPSS Statistics v.23.0 software (IBM Corp., Armonk, NY, USA) or R (https://www.r-project.org/, accessed on 17 December 2022). Data were expressed as mean ± standard deviation. In the cell viability assay, dose-dependent effects of the agent were calculated as relative scores of the vehicle control. In real-time RT-PCR, the effects of the agent on target gene/GAPDH expression were shown as relative values of the vehicle control. In the positivity analysis of immunofluorescence and protein expression measurements of western blotting, the effects of the agent were tested using replicates from the same donors. One-way repeated measures analysis of variance (ANOVA) with the TukeyeKramer post hoc test was thus used. Normal distributions were evaluated by using Shapiro–Wilk test, and the equalities were evaluated by using Bartlett’s test. Consequently, histological scores were not normally distributed. In this analysis, non-parametric Kruskal–Wallis test with the Steele–Dwass post hoc test was applied.

## 5. Conclusions

AdipoRon has the potential to decelerate the progression of IVD degeneration and contribute to a new treatment strategy against the early stage of IVD degeneration.

## Figures and Tables

**Figure 1 ijms-24-08566-f001:**
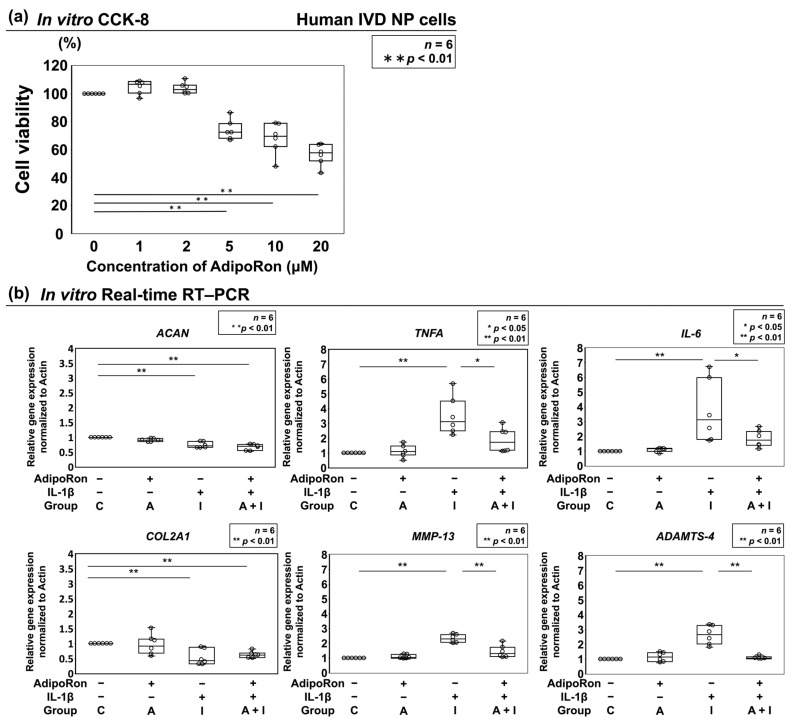
(**a**) Viability of human disc nucleus pulposus (NP) cells assessed using cell counting kit-8 (CCK-8) after 24 h treatment of 0–20 μM AdipoRon. Changes in CCK-8 dehydrogenase activity by drug treatment relative to that of the vehicle control are shown as dot and box plots. One-way repeated-measures analysis of variance (ANOVA) with the Tukey–Kramer post hoc test was used (*n* = 6). (**b**) Relative gene expression in human intervertebral disc (IVD) NP cells. Effects of AdipoRon treatment on the expression of anabolic *ACAN* encoding aggrecan, *COL2A1* encoding collagen type II, proinflammatory *TNFA* encoding tumor necrosis factor-α (TNF-α), *interleukin* (*IL*)-*6*, catabolic *matrix metalloproteinases* (*MMP*)-*13*, and *disintegrin and metalloproteinase with thrombospondin motifs* (*ADAMTS*)-*4* in IVD NP cells. NP were seeded on Tapered Stencil for Cluster Cultures (TASCL) and treated with solvent as a control (group C; set as 1.0), 2 μM AdipoRon (group A), 10 ng/mL IL-1β (group I), or both (group A + I) for 48 h. Expression of *ACAN*, *COL2A1*, *TNFA*, *IL-6*, *MMP-13*, and *ADAMTS-4* relative to that of *glyceraldehyde 3-phosphate dehydrogenase* (*GAPDH*) was assessed through real-time reverse transcription–polymerase chain reaction using SYBR Green. Data are presented as dot and box plots as fold change relative to the control values (*n* = 6). One-way repeated-measures ANOVA with the Tukey–Kramer post hoc test was used. Significant differences were set as * *p* < 0.05 and ** *p* < 0.01.

**Figure 2 ijms-24-08566-f002:**
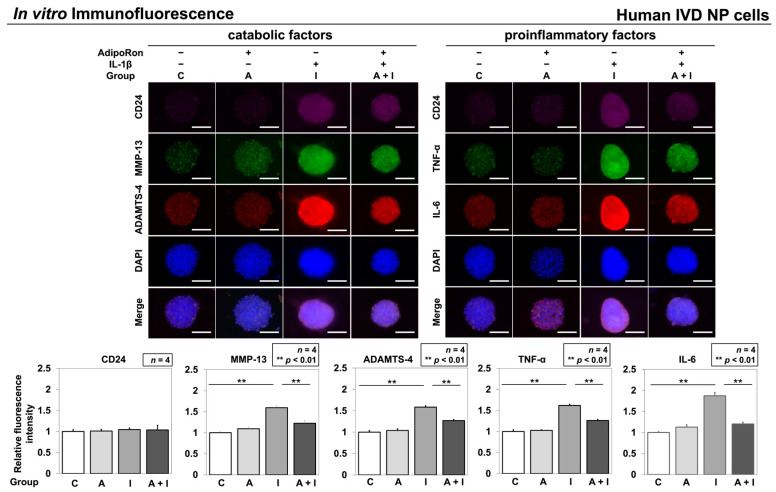
Effects of AdipoRon on CD24, ADAMTS-4, MMP-13, TNF-α, and IL-6 expression in human IVD NP cells (*n* = 4). Cells were cultured in a monolayer medium, followed by seeding at 1.0 × 10^6^ cells/TASCL to form cell clusters. After 48 h, cell clusters were treated with solvent as a control (group C, set as 1.0), AdipoRon (2 μM, group A), IL-1β (10 ng/mL, group I), or both AdipoRon and IL-1β (group A + I). Fluorescent signals of CD24 (purple), MMP-13 (green), ADAMTS-4 (red), TNF-α (green), IL-6 (red), and nuclear 4′,6-diamidino-2-phenylindole (DAPI) (blue) were captured using a fluorescence microscope. Relative fluorescence intensities of 10 randomly selected cell clusters per image were measured. Data are presented as mean ± standard deviation (*n* = 4). One-way ANOVA with the Tukey–Kramer post hoc test was used. Significant differences were set as ** *p* < 0.01. Bar = 100 μm, respectively.

**Figure 3 ijms-24-08566-f003:**
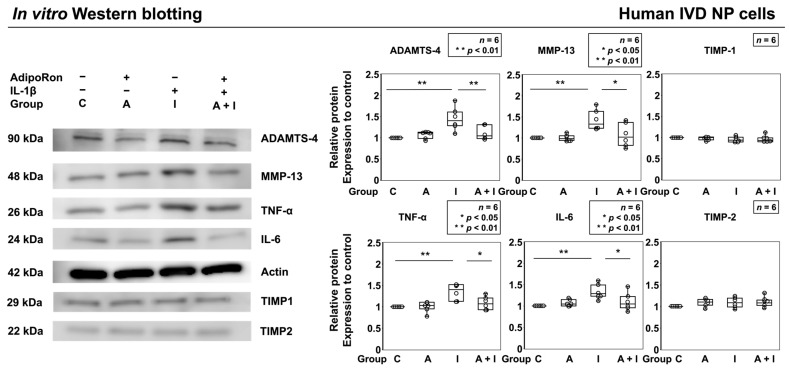
Effects of AdipoRon on ADAMTS-4, MMP-13, TNF-α, and IL-6 expression in human IVD NP cells (*n* = 6). Cells were cultured in a monolayer medium, followed by seeding at 1.0 × 10^6^ cells/TASCL to form cell clusters. After 48 h, the clusters were treated with solvent as a control (group C, set as 1.0), AdipoRon (2 μM, group A), IL-1β (10 ng/mL, group I), or both AdipoRon and IL-1β (group A + I). Total protein extracts from cells were analyzed by western blotting for extracellular matrix (ECM) catabolic factors, tissue inhibitors of matrix metalloproteinases (TIMP) and proinflammatory cytokines. Actin was used as a loading control. Immunoblots show randomly selected samples (*n* = 6). Data are presented as dot and box plots. One-way ANOVA with the Tukey–Kramer post hoc test was used. Significant differences were set as * *p* < 0.05 and ** *p* < 0.01.

**Figure 4 ijms-24-08566-f004:**
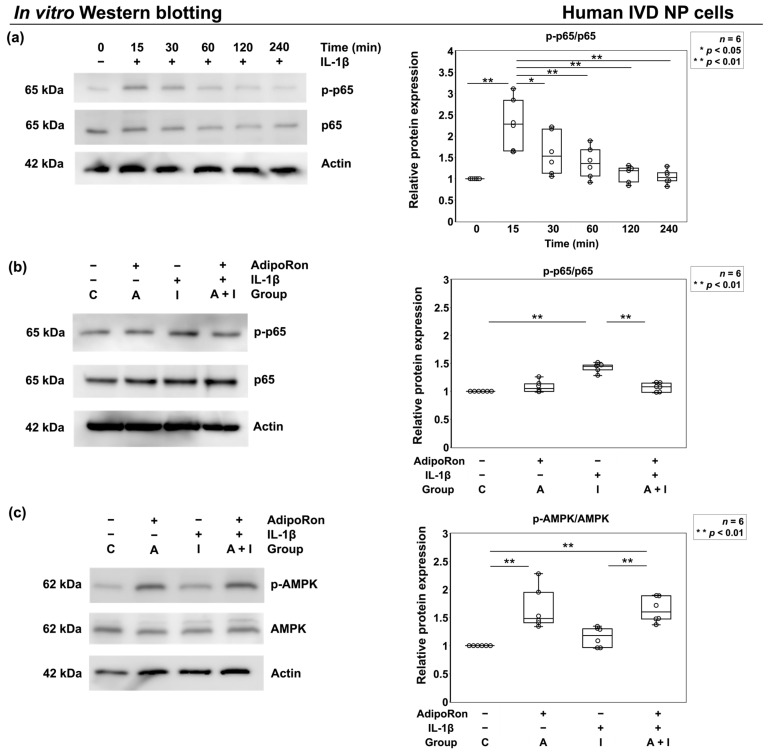
Effects of AdipoRon on p65 and adenosine monophosphate-activated protein kinase (AMPK) phosphorylation under proinflammatory conditions. (**a**) Time-course changes in phos-phorylation of p65 (p-p65) relative to p65 level. Human IVD NP cells were stimulated with 10 ng/mL IL-1β, and the levels of p-p65 and p65 were assessed at different time points, i.e., 0 (set as 1.0), 15, 30, 60, 120, and 240 min. (**b**,**c**) Expression of p-p65 relative to that of p65 and phosphorylated AMPK (p-AMPK) relative to that of AMPK in IVD NP cells treated with phosphate-buffered saline (PBS) (group C; set as 1.0), AdipoRon (2 μM, group A), IL-1β (10 ng/mL, group I), or both (group A + I) for 15 min. Data are presented as dot and box plots. Results from six independent experiments were analyzed using one-way repeated-measures ANOVA with the Tukey–Kramer post hoc test (*n* = 6). Significant differences were set as * *p* < 0.05 and ** *p* < 0.01.

**Figure 5 ijms-24-08566-f005:**
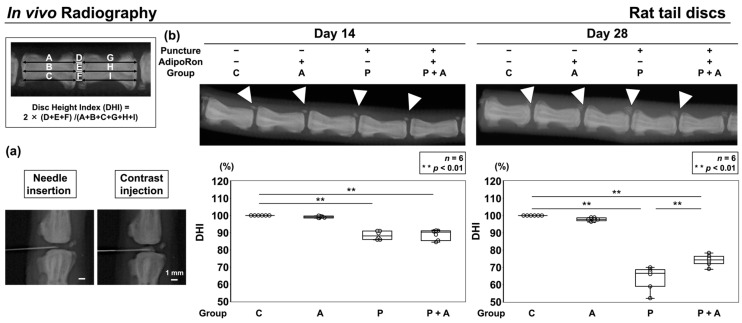
(**a**) Confirmation with radiography of successful intradiscal 33 G needle insertion and 2 μL contrast injection at the center of IVD. Bar = 1 mm, respectively. (**b**) Effects of AdipoRon on radiographic disc height in the annular puncture model of rat tail IVD. A 20-gauge (G) needle was used to puncture the caudal disc of rat at a depth of 5 mm from the skin to the center of the NP. The needle was rotated 360° and held in that position for 30 s before extraction to establish an IVD model. Following the initial puncture, either vehicle (PBS; 2 μL per disc, group P) or AdipoRon (2 μM in 2 μL per disc, group P + A) was injected. Lateral radiographs of the rat tail were obtained on postoperative days 14 and 28, and the disc height index (DHI) was calculated by measuring, averaging, and normalizing the disc height with respect to the adjacent vertebral body heights in the anterior, middle, and posterior regions. The control was without puncture and was designated as 100% (group C), and group A was without puncture and was injected AdipoRon (2 μM in 2 μL per disc) using a 33 G needle at the disc center through a 5-mm longitudinal skin incision. Then, the four groups (C, A, P, and P + A) were compared for DHI. Data are presented as dot and box plots (*n* = 6). One-way ANOVA with the Tukey–Kramer post hoc test was used. Significant differences were set as ** *p* < 0.01.

**Figure 6 ijms-24-08566-f006:**
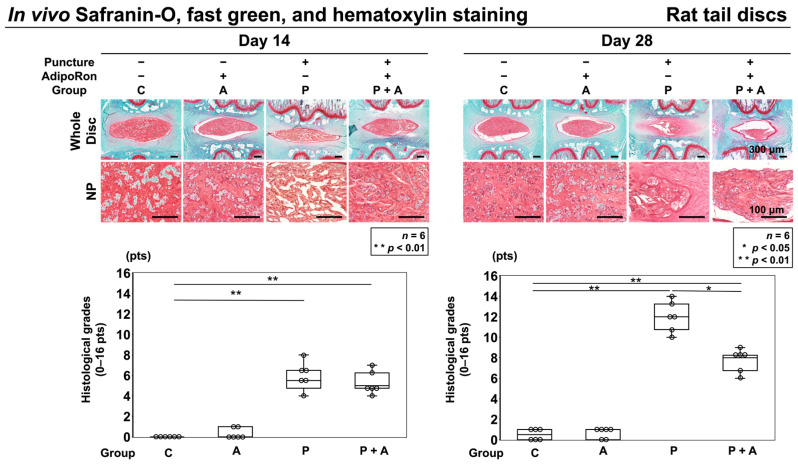
Effects of AdipoRon on histological IVD degeneration in a rat model of tail puncture. After establishing the model, either vehicle (PBS; 2 μL per disc, group P) or AdipoRon (2 μM in 2 μL per disc, group P + A) was injected. IVD degeneration was assessed by being stained with safranin-O and fast green of the IVDs at 14 and 28 days after injection in low-power fields (×100). The control was without puncture (group C), and group A was without puncture and injected AdipoRon (2 μM in 2 μL per disc) using a 33 G needle at the disc center through a 5 mm longitudinal skin incision. The four groups (C, A, P, and P + A) were compared for IVD degeneration. Data are presented as dot and box plots (*n* = 6). Kruskal–Wallis test with the Steele–Dwass post hoc test was used. Significant differences were set as * *p* < 0.05, ** *p* < 0.01. Bar = 100 μm and 300μm, respectively.

**Figure 7 ijms-24-08566-f007:**
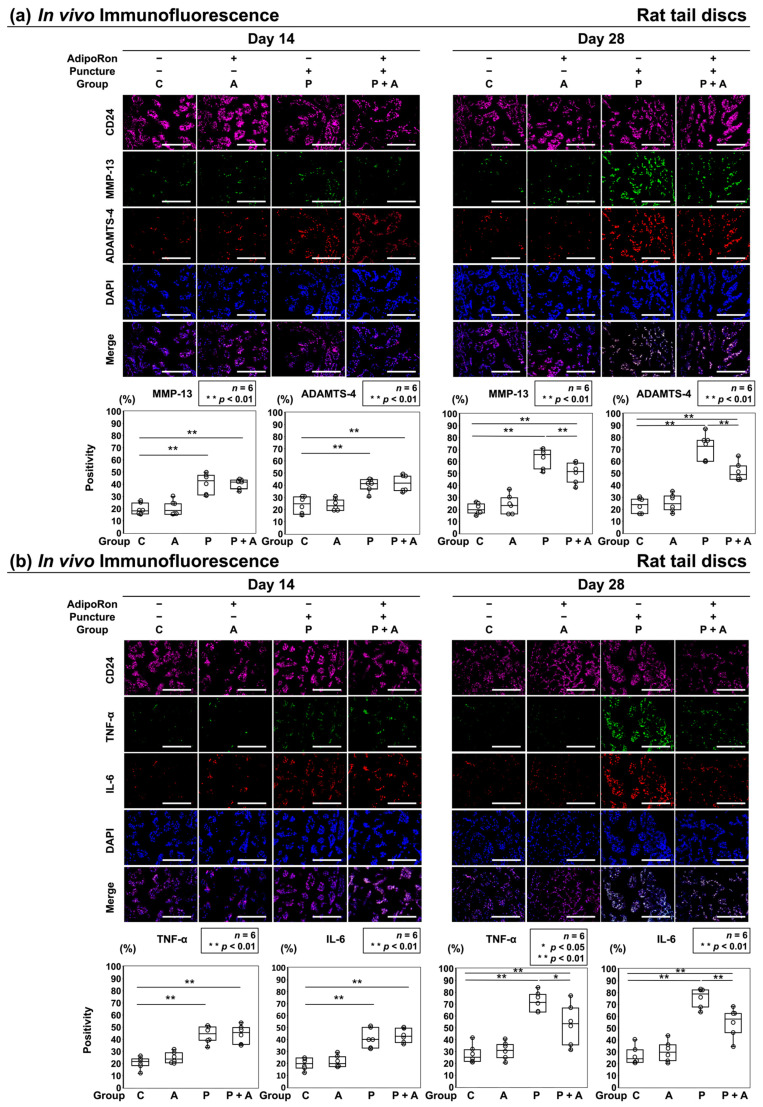
(**a**) Effects of AdipoRon on ECM catabolic factors in a rat model of tail puncture. After establishing the IVD model, vehicle (PBS; 2 μL per disc, group P) or AdipoRon (2 μM in 2 μL per disc, group P + A) was injected. ECM components were assessed using immunofluorescence (MMP-13, green; ADAMTS-4, red; CD24, purple; DAPI, blue; and merged signals) of the discs 14 and 28 days post-injection in high-power fields (400×). Immunopositivity was calculated as a percentage relative to DAPI-positive cells. The control was without puncture (group C), and group A was without puncture and injected AdipoRon (2 μM in 2 μL per disc) using a 33 G needle at the disc center through a 5 mm longitudinal skin incision. Data of the four groups (C, A, P, P + A) were compared. Data are presented as dot and box plots (*n* = 6). One-way ANOVA with the Tukey–Kramer post hoc test was used. Significant differences were set as ** *p* < 0.01. Bar = 100 μm, respectively. (**b**) Effects of AdipoRon on pro-inflammatory factors in a rat model of tail puncture. After establishing the IVD model, vehicle (PBS; 2 μL per disc, group P) or AdipoRon (2 μM in 2 μL per disc, group P + A) was injected. ECM components were assessed using immunofluorescence (TNF-α, green; IL-6, red; CD24, purple; DAPI, blue; and merged signals) of the discs 14- and 28-days post-injection in high-power fields (400×). Immunopositivity was calculated as a percentage relative to DAPI-positive cells. The control was without puncture (group C), and group A was without puncture and was injected AdipoRon (2 μM in 2 μL per disc) using a 33 G needle at the disc center through a 5 mm longitudinal skin incision. Data of the four groups (C, A, P, P + A) were compared. Data are presented as dot and box plots (*n* = 6). One-way ANOVA with the Tukey–Kramer post hoc test was used. Significant differences were set as * *p* < 0.05, ** *p* < 0.01. Bar = 100 μm, respectively.

**Figure 8 ijms-24-08566-f008:**
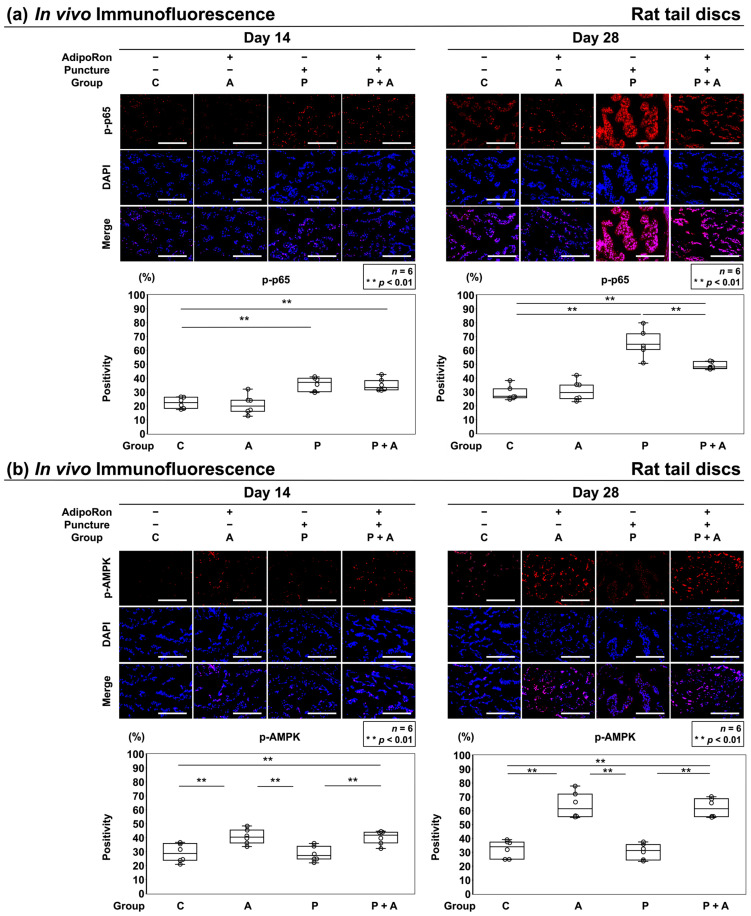
(**a**) Effects of AdipoRon on p-p65 expression in a rat model of tail puncture. After establishing the IVD degeneration model, vehicle (PBS; 2 μL per disc, group P) or AdipoRon (2 μM in 2 μL per disc, group P + A) was injected. Proinflammatory cytokines were assessed using immunofluorescence (p-p65, red; DAPI, blue; and merged signals) of the IVDs at 14 and 28 days after injection in high-power fields (400×). Immunopositivity was calculated as a percentage relative to DAPI-positive cells. The control was without puncture (group C), and group A was without puncture and injected AdipoRon (2 μM in 2 μL per disc) using a 33 G needle at the disc center through a 5 mm longitudinal skin incision. The four groups (C, A, P, P + A) were compared for immunopositivity. Data are presented as dot and box plots. One-way ANOVA with the Tukey–Kramer post hoc test was used (*n* = 6). Significant differences were set as ** *p* < 0.01. Bar = 100 μm, respectively. (**b**) Effects of AdipoRon on AMPK phosphorylation (p-AMPK) in a rat model of tail puncture. After establishing the IVD degeneration model, vehicle (PBS; 2 μL per disc, group P) or AdipoRon (2 μM in 2 μL per disc, group P + A) was injected. AMPK pathways were assessed using immunofluorescence (p-AMPK, red; DAPI, blue; and merged signals) of the IVDs at 14 and 28 days after injection in high-power fields (400×). Immunopositivity was calculated as a percentage relative to DAPI-positive cells. The control was without puncture (group C), and group A was without puncture and was injected AdipoRon (2 μM in 2 μL per disc) using a 33 G needle at the disc center through a 5 mm longitudinal skin incision. The four groups (C, A, P, P + A) were compared for immunopositivity. Data are presented as dot and box plots. One-way ANOVA with the Tukey–Kramer post hoc test was used (*n* = 6). Significant differences were set as ** *p* < 0.01. Bar = 100 μm, respectively.

**Figure 9 ijms-24-08566-f009:**
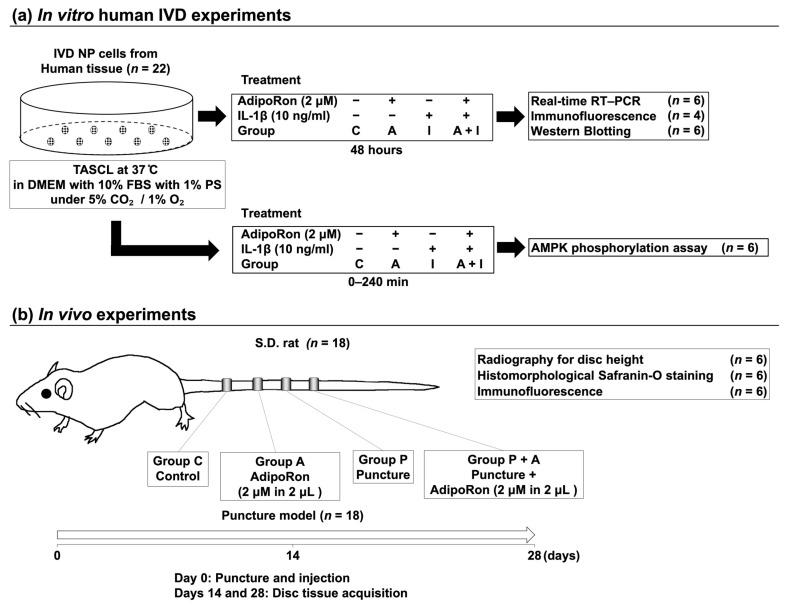
(**a**) Schematic illustration of in vitro experimental regimens. In the first passage, approximately 80% confluent human IVD NP cells from surgical waste of patients with degenerative lumbar spine disease (total *n* = 22) were cultured as a monolayer followed by seeding into a three-dimensional culture system (TASCL) to form cell clusters. Cell clusters were treated with AdipoRon (2 μM, group A), IL-1β (10 ng/mL, group I), or both (group A + I). Non-treated cell clusters were used as the control (group C). Then, cell clusters were analyzed using real-time reverse transcription–polymerase chain reaction (RT–PCR), immunofluorescence, and western blotting. (**b**) Schematic illustration of in vivo experimental regimens. Annular punctures at C10−C11 and C11−C12 discs were performed in the tails of 12-week-old male Sprague–Dawley rats (*n* = 18), while the C8−C9 disc was later collected as the non-treated control (groups C). At C9−C10, 2 μL AdipoRon was injected using a 33-gauge needle at the disc center through a 5 mm longitudinal skin incision (2 μM in 2 μL per disc, group A). Immediately following the initial puncture at C10−C11 and C11−C12, AdipoRon (2 μM in 2 μL per disc, group P + A) was injected into the center of the IVD NP, and non-injected samples were in group P. Disc height evaluation, histomorphology, and immunofluorescence were performed 14 and 28 days after the puncture (*n* = 6/time point).

**Table 1 ijms-24-08566-t001:** Primer sequences used in this study.

Gene Name	Forward (5′–3′)	Reverse (5′–3′)
*ACAN*	GTCAGATACCCCATCCACACTC	CATAAAAGACCTCACCCTCCAT
*COL2A1*	CTCAAGTCGCTGAACAACCA	GTCTCCGCTCTTCCACTCTG
*TNFA*	AGGCGGTGCTTGTTCCTCA	GTTCGAGAAGATGATCTGACTGCC
*IL-6*	ATGAACTCCTTCTCCACAAGCGC	GAAGAGCCCTCAGGCTGGACTG
*MMP-13*	CCAGGCATCACCATTCAAG	ATCATCTTCATCACCACCACTG
*ADAMTS-4*	CCTGGCAAGGACTATGATGCTGA	GGGCGAGTGTTTGGTCTGG
*GAPDH*	GTTCGACAGTCAGCCGCAT	GGAATTTGCCATGGGTGGA

*ACAN* = aggrecan; *ADAMTS* = a disintegrin and metalloproteinase with thrombospondin motifs; *COL2A1* = collagen type II alpha 1 chain; *GAPDH* = glyceraldehyde-3-phosphate dehydrogenase; *IL* = interleukin; *MMP* = matrix metalloproteinase; *TNFA* = tumor necrosis factor-α.

## Data Availability

Not applicable.

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
