# Peer review of "Anti-Inflammatory Effects of Adiponectin Receptor Agonist AdipoRon against Intervertebral Disc Degeneration"

_ijms, 2023, doi:10.3390/ijms24108566_

Round 1

Reviewer 1 Report

Dear authors, thanks for your effort in writing the present manuscript.
Over all, the manuscript is clear, well written and with a good architecture.

In order to complete the information present on the discussion section, this reviewer would add a sentence on mechanobiology of the disc as one of the causes of IVD degeneration, with consequent reference (eg, Mechanobiology of the Human Intervertebral Disc: Systematic Review of the Literature and Future Perspectives. Int J Mol Sci. 2023 Feb ).

In conclusion, I advise acceptance after minor revision of the manuscript

Author Response

Reviewer #1

Comments and Suggestions for Authors

Dear authors, thanks for your effort in writing the present manuscript. Over all, the manuscript is clear, well written and with a good architecture.

Authors’ response: Thank you for your expertise. We revised the manuscript based on your comments.

In order to complete the information present on the discussion section, this reviewer would add a sentence on mechanobiology of the disc as one of the causes of IVD degeneration, with consequent reference (eg, Mechanobiology of the Human Intervertebral Disc: Systematic Review of the Literature and Future Perspectives. Int J Mol Sci. 2023 Feb).

Authors’ response: Thank you for your advice. We included details in the discussion (Lines 541–544).

Reviewer 2 Report

The manuscript focuses on the influence of AdipoRon on IVD inflammatory and metabolic factor expressions. AdipoRon is indeed a promising and innovative treatment of IVD degeneration. In the manuscript, the data are well presented. There are some suggestions enclosed as follows:

The reason of choosing MMP-13 and ADAMTS-4 for evaluation may need to be justified, since upregulation of other MMPs (MMP-1, -2, -3, -7, -8, -10) and ADAMTS (ADAMTS-1, and -15) were also identified in human degenerated IVDs. (PMID: 23369495)

MMP13 expression may not completely represent catabolism of matrix. Activity of MMPs may be better evaluated using gelatin zymography or other MMP activity assays. MMP13 activity is also influenced by inhibitors of metalloproteinases (TIMPs).

It is a great strategy to validate in vitro findings with in vivo model. However, notice that in vitro NP cells were stimulated by IL-1β, while in vivo the IVD degeneration is caused by puncture. Is there association between puncture and IL-1β protein release?

Why AdipoRon and PBS control were delivered using 27G needle after puncture, which is different from the 33G needle delivering AdipoRon without puncture? Why not to use 33G to deliver AdipoRon/PBS for all these groups?

The endplates are not straight lines, how was the disc height measured?

In the method part of in vivo testing, how the randomization was performed? Which strategy was used to make sure the puncture was ''at a depth of 5 mm'' and the drug delivery was ''at the center of IVD''?

For immunofluorescent staining of paraffin-embedded tissue sections, which antigen retrieval method was used? How the location of imaging decided? How many images per sample were included? Are they averaged by mean or median?

Are P+A group always at C11-12 level, and P group always at C10-11 level? What has been explained is that C9-10 and C8-9 are always used for 2 different controls. Would the result be confounded by the IVD levels? For example, is there a possibility that C10-11 shows difference compared to C8-9 not because of different treatments (puncture or not), but because these 2 levels are anyway different?

Would encourage to show data points of donor levels for all the plots. It would be very informative to observe the differences among donors. For statistic, which method is used to test normality and equality of variance, since ANOVA is used?

The conclusion has introduced the concept of ''IVD degenerative disease''? Please clarify its definition. Should an IVD degeneration without symptom regarded as ''IVD degenerative disease''? Does upregulation of certain genes and activation of certain cellular pathway represent a ''disease'' with symptoms? Is the rodent model testing enough to justify the AdipoRon injection as ''a therapeutic candidate''? If yes, what is the evidence of its safety and efficiency in treating low back pain when translating to human?

Author Response

RESPONSES TO THE REVIEWERS’ COMMENTS

Reviewer #2

Comments and Suggestions for Authors

The manuscript focuses on the influence of AdipoRon on IVD inflammatory and metabolic factor expressions. AdipoRon is indeed a promising and innovative treatment of IVD degeneration. In the manuscript, the data are well presented. There are some suggestions enclosed as follows:

Authors’ response: Thank you for your expertise. We revised the manuscript based on your comments.

The reason of choosing MMP-13 and ADAMTS-4 for evaluation may need to be justified, since upregulation of other MMPs (MMP-1, -2, -3, -7, -8, -10) and ADAMTS (ADAMTS-1, and -15) were also identified in human degenerated IVDs. (PMID: 23369495)

Authors’ response: Thank you for pointing this out. As you pointed out, it is known that various ADAMTS and MMPs are expressed in intervertebral disc degeneration. We have chosen ADAMTS-4 and MMP-13, which have been studied more as representative extracellular matrix catabolic factors for the target of treating intervertebral disc degeneration. We additionally cited references;

(1)       Huang, Y. et al. (2021). " circSPG21 protects against intervertebral disc disease by targeting miR-1197/ATP1B3" Exp Mol Med. 53(10): 1547–1558

(2)       Cheng, X. et al. (2018). " Circular RNA VMA21 protects against intervertebral disc degeneration through targeting miR-200c and X linked inhibitor-of-apoptosis protein" Ann Rheum Dis. 77(5): 770-779

(Lines 122–124)

MMP13 expression may not completely represent catabolism of matrix. Activity of MMPs may be better evaluated using gelatin zymography or other MMP activity assays. MMP13 activity is also influenced by inhibitors of metalloproteinases (TIMPs).

Authors’ response: Thank you for pointing this out. We additionally evaluated TIMP-1 and 2 in Western blotting (Figure 3) and added a discussion of TIMPs.

(Lines 605–606)

It is a great strategy to validate in vitro findings with in vivo model. However, notice that in vitro NP cells were stimulated by IL-1β, while in vivo the IVD degeneration is caused by puncture. Is there association between puncture and IL-1β protein release?

Authors’ response: Thank you for pointing this out. Since several papers have reported that the rat puncture model produces an inflammatory response, we consider this technique a proper way to induce inflammation in place of IL-1β stimulation in vivo. We mentioned it in Results section (Lines 210–212).

Why AdipoRon and PBS control were delivered using 27G needle after puncture, which is different from the 33G needle delivering AdipoRon without puncture? Why not to use 33G to deliver AdipoRon/PBS for all these groups?

Authors’ response: I’m sorry that this part was not clear in the original manuscript.  With the 20G needle punctured, the 33G needle is too short to reach the center of the intervertebral disc through the 20G needle because it hits the connection of the 20G needle (the length of each needle: 27G 47 mm, 33G 14mm). Therefore, the punctured group used a 27G needle, which can pass through 20G and is long enough. We included details in the text (Lines 799–800).

The endplates are not straight lines, how was the disc height measured?

Authors’ response: I’m sorry that this part was not clear in the original manuscript.

DHI was calculated by measuring, averaging, and normalizing the disc height against the height of adjacent vertebrae in the anterior, middle, and posterior regions to correct the asymmetry form of each intervertebral disc. We included details in the text (Lines 810–812).

In the method part of in vivo testing, how the randomization was performed? Which strategy was used to make sure the puncture was ''at a depth of 5 mm'' and the drug delivery was ''at the center of IVD''?

Authors’ response: Thank you for pointing that out. As you questioned, the word 'randomly' was inappropriate, as we had set up the groups for each vertebrae level (eg. C8-9, group C; C9-10, group A; C10-11, group P; C11-12, group P+A). We apologize for the incorrect wording and correct it. In accordance with previous studies, we determined this disc puncture method. We included details in the text (Lines 793–797). Then, we performed the additional experiment using this technique with a contrast agent administered to the intervertebral disc, showing that it was delivered exactly at the center of IVD. (figure5)

For immunofluorescent staining of paraffin-embedded tissue sections, which antigen retrieval method was used? How the location of imaging decided? How many images per sample were included? Are they averaged by mean or median?

Authors’ response: I’m sorry that this part was not clear in the original manuscript. I used Heat-Induced Epitope Retrieval for the antigen retrieval method. Four random locations per sample were taken and their average positivity was calculated. We included details in the text (Lines 835, 847).

Are P+A group always at C11-12 level, and P group always at C10-11 level? What has been explained is that C9-10 and C8-9 are always used for 2 different controls. Would the result be confounded by the IVD levels? For example, is there a possibility that C10-11 shows difference compared to C8-9 not because of different treatments (puncture or not), but because these 2 levels are anyway different?

Authors’ response: Thank you for pointing this out. We set P+A group at C11-12 level and P group at C10-11 level. Although your suggested possibility cannot be ruled out, we did not find any studies which examined if the differences in the level of rat caudal spine affected the degree of disc degeneration.

Would encourage to show data points of donor levels for all the plots. It would be very informative to observe the differences among donors. For statistic, which method is used to test normality and equality of variance, since ANOVA is used?

Authors’ response: Thank you for pointing this out. We arranged the figures showing data points of donor levels with data re-collection and statistical analysis performed again. Based on feedback from additional experiments and minor corrections from data re-collection, we have made some corrections to the data, but they do not affect the results (Figure1,3–8) We used to test the normality by Shapiro-Wilk normality test and the equality by Bartlett’s test. For Figure 6, it was necessary to change to the non-parametric KruskaleWallis test with the SteeleDwass posthoc test, but this did not affect the results. We included details in the text (Lines 850–863).

The conclusion has introduced the concept of ''IVD degenerative disease''? Please clarify its definition. Should an IVD degeneration without symptom regarded as ''IVD degenerative disease''? Does upregulation of certain genes and activation of certain cellular pathway represent a ''disease'' with symptoms? Is the rodent model testing enough to justify the AdipoRon injection as ''a therapeutic candidate''? If yes, what is the evidence of its safety and efficiency in treating low back pain when translating to human?

Authors’ response: I’m sorry that this part was not clear in the original manuscript. IVD degenerative disease would be a disease with symptoms, such as low back pain or sciatica, caused by IVD degeneration. So, if people with IVD degeneration do not have any symptoms, they are not considered to have IVD degenerative disease. Upregulation of certain genes and activation of certain cellular pathways would introduce more degenerative changes in IVD. People with more degenerated IVD will have more risk to have IVD degenerative disease. Therefore, we consider that the prevention of IVD degeneration would result in the prevention of IVD degenerative disease. However, our data could not show evidence that AdipoRon could improve low back pain or treat IVD degenerative disease. So we changed the manuscript. (Lines 642–643, 865–866).

Round 2

Reviewer 2 Report

The revised manuscript has addressed most of the comments. As already mentioned in the first round of revision, it will be of great scientific value if the methodology can be more detailed. Such as the 'heat-induced epitope retrieval' needs more description. Would suggest to add the heating method and duration. Also, for the injection, how to inject into IVD center with high consistently and without contrast agent assistance? These details will be informative for future studies in this direction and can improve the manuscript's impact.

As admitted that the confounding effect of spinal segments is unknown, may have this issue discussed in the discussion session. The data cannot support that ''AdipoRon could be an excellent therapeutic candidate for treating degenerative IVD disease'' (in the abstract and also in conclusion), but may suggest a role in alleviating early stage IVD degeneration. Whether AdipoRon treat degenerative IVD disease or not needs evidence of symptom improvement.

Author Response

RESPONSES TO THE REVIEWERS’ COMMENTS

Reviewer #2

The revised manuscript has addressed most of the comments. As already mentioned in the first round of revision, it will be of great scientific value if the methodology can be more detailed.

Authors’ response: Thank you for your expertise. We revised the manuscript based on your comments again.

Such as the 'heat-induced epitope retrieval' needs more description. Would suggest to add the heating method and duration. Also, for the injection, how to inject into IVD center with high consistently and without contrast agent assistance? These details will be informative for future studies in this direction and can improve the manuscript's impact.

Authors’ response: I’m sorry that this part was not clear in the original manuscript.  We have modified the Methods part about the immunofluorescence in vivo and the rat procedures. We included details in the text (Lines 559–561, 601–603).

As admitted that the confounding effect of spinal segments is unknown, may have this issue discussed in the discussion session. The data cannot support that ''AdipoRon could be an excellent therapeutic candidate for treating degenerative IVD disease'' (in the abstract and also in conclusion), but may suggest a role in alleviating early stage IVD degeneration. Whether AdipoRon treat degenerative IVD disease or not needs evidence of symptom improvement.

Authors’ response: Thank you for pointing this out. We totally agreed with you. So, we included details in the text (Lines 28–29, 401–402, 630–631).